# RoboKoop: Efficient Control Conditioned Representations from Visual Input in Robotics using Koopman Operator

**Hemant Kumawat, Biswadeep Chakraborty, and Saibal Mukhopadhyay**
Georgia Institute of Technology
{hkumawat6, biswadeep, smukhopadhyay6} @gatech.edu

**Abstract:** Developing agents that can perform complex control tasks from high-dimensional observations is a core ability of autonomous agents that requires underlying robust task control policies and adapting the underlying visual representations to the task. Most existing policies need a lot of training samples and treat this problem from the lens of two-stage learning with a controller learned on top of pre-trained vision models. We approach this problem from the lens of Koopman theory and learn visual representations from robotic agents conditioned on specific downstream tasks in the context of learning stabilizing control for the agent. We introduce a Contrastive Spectral Koopman Embedding network that allows us to learn efficient linearized visual representations from the agent's visual data in a high dimensional latent space and utilizes reinforcement learning to perform off-policy control on top of the extracted representations with a linear controller. Our method enhances stability and control in gradient dynamics over time, significantly outperforming existing approaches by improving efficiency and accuracy in learning task policies over extended horizons.

**Keywords:** Feature extraction, Task Feedback, Koopman Operator

## 1 Introduction

Building agents capable of executing intricate control tasks using high-dimensional inputs, like pixels is crucial in many real-world applications. Combining deep learning alongside reinforcement learning [1, 2, 3], various methods have been developed for learning representations from visual data in robotics for executing intricate control tasks. Most of these learning algorithms can be fundamentally organized into two interconnected yet distinct research directions (refer to Figure 1 and Table 1): (1) Developing visual predictive models that learn the underlying dynamics [4, 5, 6, 7] in latent space by predicting the future visual observations. (Figure 1-A ) (2) Using deep RL to perform tasks with non-linear modeling in an interactive environment[8, 9, 10] (Figure 1-B). The first approach involves creating predictive models of the environment which are then leveraged for tasks or generating samples, which model-free methods can then use for learning. However, these models could have sub-optimal task performance. They may need either task-visual alignment [11, 12, 13] where pre-trained vision model's parameters are adapted to the task via end-to-end training or manual task and control parameters tuning to achieve good task performance. Alternatively, RL-based methods learn the representation by self-supervised auxiliary tasks but they may need a huge amount of interactions with the environment as the dynamics are learned implicitly in the form of policy networks. In general, all these approaches are not *sample efficient* i.e., they need to collect a large number of samples from environment interactions to achieve satisfactory task performance. Additionally, these models often use computationally expensive latent dynamics models like transformers, MLP, and RNN.

In our work, we aim to learn task-conditioned representations (Figure 1-C) from visual observations in a sample-efficient way without sacrificing task accuracy by modeling the dynamics of the representations linearly. An important advantage of linear systems is their generalized math-

8th Conference on Robot Learning (CoRL 2024), Munich, Germany.

ematical frameworks, in contrast to nonlinear systems which have no overarching mathematical frameworks for general characterization of systems. Linear systems are very well defined by their spectral decomposition, enabling the development of generalizable and efficient control algorithms.

Prior works [16] have explored learning high semantic information from visual input with latent dynamics governed by dense Koopman. However such models are proven [17][18] to be computationally inefficient and require a large number of samples due to bigger parametric space with no stability guarantees of the system. In contrast, we model the dynamics of the task embedding space with spectral Koopman decomposition which is proven to learn rich linear representations for time series modeling while also being computationally efficient. We hypothesize that *if we can linearly model the task embedding space and identify a finite subset of relevant, stable (negative) eigenvalues of the Koopman operator that governs its dynamics while conditioned on task, we can learn rich agent representations for the task with lesser number of environmental interactions while also allowing us to perform efficient parallelized forward latent prediction.* Our main contributions are as follows: (a.) We propose a novel contrastive spectral Koopman encoder to map the visual input to a complex-valued task embedding space with dynamics governed by a learnable spectral Koopman operator. We then use reinforcement learning to learn this embedding space, spectral Koopman operator, and its associated linear task controller with prediction as an auxiliary task. Our network is sample efficient and has better task performance compared to prior works as shown in Table 1. (b.) We conduct a theoretical analysis and examine the convergence behavior of

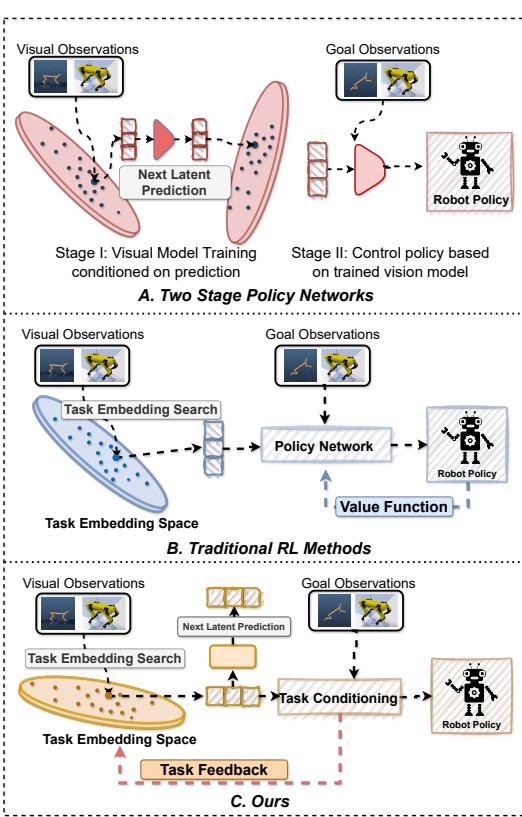

Figure 1: A. Prediction-conditioned vision models used to learn task policies for robots.[14] B. Traditional RL methods [15] C. Our method where the visual representations are conditioned on the task policy.

our method and show that our method converges to the optimal task policy given sufficient environment interactions. (c.) Through empirical tests on six simulated robotic tasks from deepmind control environment[19], we demonstrate our model's superior performance and sample efficiency against nonlinear and state dynamical models while being robust to sensory errors and external disturbances.

Table 1: Evaluation of Task Performance in Reinforcement Learning: Linear vs. Non-Linear Models. Task performance is visually indicated by arrows, with green up arrows denoting superior performance and red down arrows indicating inferior performance. The extent of performance variation, either improvement or degradation, is qualitatively represented by the number of arrows.

| Model | Dynamics Model Type | Prior Work | Sample Efficiency | Task Performance | Robust to External Disturbances | Theoretical Analysis |
|---|---|---|---|---|---|---|
| **Vision Predictive Model** | Linear Models: Dense Koopman | [17, 20, 21, 22, 23] | ✗ | ↑ | ✓ | ✗ |
| | Non-Linear Models: MLP, Lagrangian, Hamiltonian | [7, 24, 25, 26, 27, 28] | ✗ | ↓↓↓ | ✗ | ✗ |
| **RL Models** | Non-Linear: MLP, Transformer, GRU | [29, 30, 31, 9] | ✗ | ↑ | ✓ | ✗ |
| **Task Conditioned Models** | Dense Koopman | [16] | ✗ | ↓↓↓ | ✗ | ✗ |
| | Spectral Koopman | **Ours*** | ✓ | ↑↑↑ | ✓ | ✓ |

## 2  Background

**Contrastive Learning**: Contrastive Learning has emerged as a powerful framework for learning representations that capture similarity constraints within datasets when there are no data labels. It operates on a principle analogous to performing a search within a dictionary, wherein the task involves identifying matches and mismatches—akin to positive and negative pairs—as if they were keys to a specific query or anchor point [32, 33, 9, 34, 35]. Methods like Curl [9] and To-KPM [16] employ contrastive encoders to directly learn representations from images and control the system. However, these methods require a large number of samples (approximately 1000k) to achieve good task performance, while Curl attempts to learn the task without any explicit dynamic model [9].

**Dynamical Models**: Prior works model system dynamics from visual observations through deep learning networks, mapping high-dimensional observation spaces to latent embeddings. These networks learn latent embedding evolution using methods like MLP[7], RNNs[36] or neural ODE[37, 38], enabling accurate future state predictions by extracting relevant features and understanding state relations. Methods like HNN [28] employ Hamiltonian dynamics, while Lag-VAE [24, 25] utilize Lagrangian dynamics for long-term prediction and control. However, these approaches often face low prediction accuracy and require system state supervision, such as prior object segmentation [25, 39, 27, 26], to enhance Lagrangian model accuracy.

**Self-supervised learning and RL**: Multiple works [40, 41, 42] have utilized self-supervised learning to develop world predictive models, which are subsequently employed for control tasks or specific tasks. In [7], the authors learn a linearized latent space, which is then controlled using optimal control methods. However, there is no feedback from the control to guide the task in learning control-oriented features. Reinforcement Learning (RL) based methods [43, 44, 45] for directly controlling systems from pixels have witnessed significant advancements in recent years, leveraging deep learning to interpret complex visual inputs. However, one of the significant challenges in RL, especially when applied to environments with high-dimensional input spaces such as images (pixels), is sample inefficiency. This inefficiency leads to the requirement for an excessively large number of interactions with the environment for the agent to learn an effective policy.

## 3  Proposed Model

### 3.1  Problem Formulation

We consider an unknown time-invariant dynamical system of the form: $s(t+1) = \boldsymbol{F}(\boldsymbol{s}(t), \boldsymbol{u}(t)) + \boldsymbol{\xi}$ where $\boldsymbol{s}(t) \in \boldsymbol{\mathcal{S}} \subseteq \mathbb{R}^n, \boldsymbol{u}(t) \in \boldsymbol{\mathcal{U}} \subseteq \mathbb{R}^m$, and $\boldsymbol{\xi} \sim \mathcal{N}(0, \boldsymbol{\Sigma_\xi})$ are the system state, control input and system noise respectively. Function $\boldsymbol{F}(\boldsymbol{s}(t), \boldsymbol{u}(t)) : \boldsymbol{\mathcal{S}} \times \boldsymbol{\mathcal{U}} \to \boldsymbol{\mathcal{S}}$ governs the transition of the states of the dynamical system and is assumed to be arbitrary, smooth and non-linear. We assume that we can only observe the system through the visual depictions $\boldsymbol{x}(t)$. Our objective is to identify a control sequence $u_{0:T}$ that minimizes the cumulative task cost function $c(x_k, u_k)$ over $T$ time steps by learning an implicit embedding function $\boldsymbol{\Phi} : \mathcal{X} \to \mathcal{Z}$ that maps the pixel space to some latent space. Further, the evolution of latent variable $\dot{\boldsymbol{z}} = g(\boldsymbol{z}, \boldsymbol{u}), g : \mathcal{Z} \times \mathcal{U} \to \mathcal{Z}$ is parameterized by a neural network $\Lambda$. The network $\Lambda$ is constrained to follow linear dynamics with a Koopman operator $\mathcal{K}$ given by $\Phi(\boldsymbol{x}_{t+1}, \boldsymbol{u}_{t+1}) = \mathcal{K}\Phi(\boldsymbol{x}_t, \boldsymbol{u}_t) = Az_t + Bu_t$ For control problems, we formulate the task cost function as the optimal control problem in latent space given by: $\min\limits_{u_{0:T-1}} \sum_{k=0}^{T-1} c(z_k, u_z)$ subject to $z_{t+1} = Az_t + Bu_t$. We aim to learn this Koopman Operator and linear control policy $u = \pi(s)$ with the Koopman embedding function mapping the high dimensional input to latent space conditioned on the control cost of the system.

### 3.2  RoboKoop Model Design

Figure 2 presents our overall methodology for learning a parameterized, linear Koopman embedding manifold via a contrastive spectral Koopman encoder. It generates key and query samples corresponding to each observation at time $t$. Positive samples are created by applying image aug-

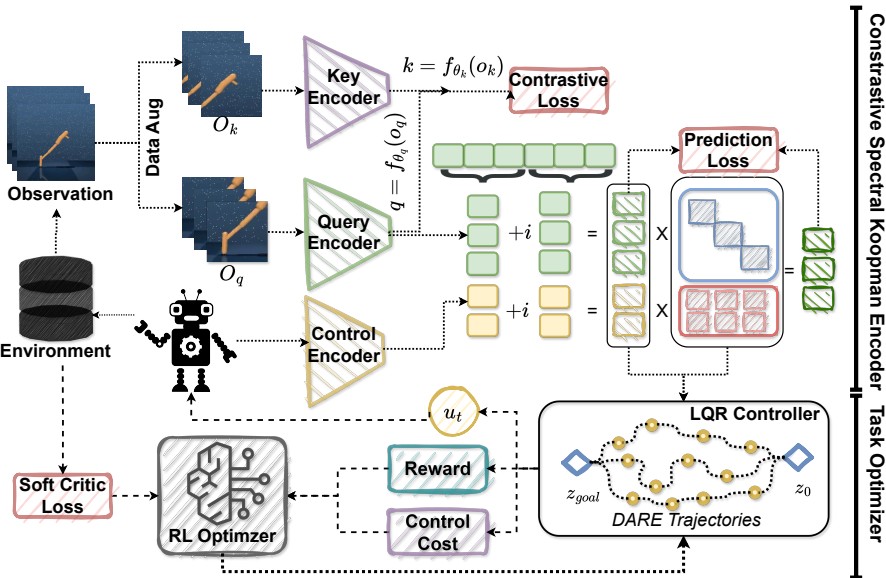

Figure 2: Our model RoboKoop with contrastive spectral Koopman encoding and RL guided control conditioning

mentations, such as random cropping [9], to the state $x_t$, whereas negative samples result from applying analogous augmentations to all other states except $x_t$. These samples are further processed by dedicated key and query encoders, which update their parameters based on the similarity measure between the samples. The output of the query encoder is then used to map the visual space to Koopman embedding space by bifurcating the query vector into real and imaginary components. This introduces a complex-valued Koopman embedding space, characterized by a spectral Koopman operator with learnable eigenvalues expressed as $\mu_i + j\omega$. Utilizing this embedding space and its spectral Koopman operator, our approach focuses on deriving optimal control strategies and their parameters through the iterative optimization of a Linear Quadratic Regulator (LQR) problem for a predefined finite time horizon, guiding the system towards a specified goal state. This goal state is also provided in visual space and subsequently mapped to the Koopman embedding space through the key encoder. Moreover, we construct value functions through dual Q-value functions, serving as critics within the Soft Actor-Critic (SAC) algorithm framework. Model updates are performed following the SAC methodology [15], grounded on the cost incurred by the LQR controller. For training, we iteratively gather data batches from the environment, applying three specific losses to train our models: 1) SAC loss for the critic and Koopman parameters, 2) Contrastive loss for optimizing contrastive encoder parameters, and 3) Next latent prediction loss for regularizing the Koopman embedding space dynamics.

**a.) Contrastive Spectral Koopman Representations** In our work, we diverge from traditional temporal modeling techniques, which typically process a sequence of consecutive frames to understand scene dynamics [46]. Instead, inspired by contrastive learning approaches [9] [16], we introduce a dual-encoder framework consisting of key and query encoders to extract nuanced visual representations. Each encoder processes high-dimensional inputs to generate corresponding sets of key and query samples. To enrich the diversity of these samples, we employ a random crop data augmentation technique on each input sequence $X = \{x_i | i = 0, 1, 2, \ldots\}$, creating positive samples $x_i^+$ and negative samples from the augmented states of $X \setminus \{x_i\}$. These samples are then embedded via the encoders to produce embeddings $z_i^q$, $z_i^+$, and $z_j^-$.

Building on this, we map the query embeddings into a complex Koopman embedding space, by splitting the query embedding vector into real and imaginary components, to model continuous latent dynamics. This is complemented by a control embedding that maps robot control inputs to a complex-valued Koopman control space. We postulate that the latent and control spaces within it are

independent, adhering to a model of continuous latent dynamics. This is mathematically represented as $\frac{dz}{dt} = Kz(t) + Lu(t)$, where the evolution of the system over time can be resolved through $z(s) = e^{Ks}z(0) + \int_0^s e^{K(s-t)}Lu(t)dt$. Here, $e^{Ks}$ denotes the matrix exponential. For practical applications, this continuous model can be converted into a discrete form as $\hat{x}_{t+1} = \bar{K}x_t + \bar{L}u_t$, achieved by applying Zero-Order Hold (ZOH) discretization to the original equations, resulting in $\bar{K} = \exp(\Delta t K)$ and $\bar{L} = (K)^{-1}(\exp(\Delta t K) - 1)L$. In scenarios where observations are uniformly sampled over time, this approach incorporates a learnable time step parameter $\Delta t$ for adjustment.

To enhance efficiency in latent space forecasting, we introduce a diagonalized Koopman matrix, $\tilde{K} = \text{diag}(\tilde{\lambda}_1, \ldots, \tilde{\lambda}_m)$, facilitating the use of an $m \times (\tau + 1)$ Vandermonde matrix $V$ [17] for straightforward row-wise circular convolutions. This method reduces the computational load by avoiding complex matrix operations.

The future latent state predictions for latent $z_t = \left[ z_t^i, \ldots, z_t^i \right]$ and control $c_t = \bar{L}u_t$ are given by

$$\left[ \hat{z}_{(t+1)}^i, \ldots, \hat{z}_{(t+\tau)}^i \right] = \left[ \tilde{\lambda}^i, \ldots, \tilde{\lambda}_i^\tau \right] z_t^i + \left[ 1, \tilde{\lambda}^i, \ldots, \tilde{\lambda}_i^{\tau-1} \right] * \left[ c_t^i, \ldots, c_{t+\tau-1}^i \right] \tag{1}$$

where $*$ denotes the circular convolution operation and $c_t$. This formulation allows for efficient computation, particularly with Fast Fourier Transform (FFT) techniques. For model training, we use a contrastive loss function, $L_{\text{cst}}$, defined as

$$L_{\text{cst}} = \mathbb{E}_{t \sim B} \left[ \log \frac{\exp(z_i^{q\top} W z_i^+)}{\exp(z_i^{q\top} W z_i^+) + \sum_{j \neq i} \exp(z_i^{q\top} W z_j^-)} \right], \tag{2}$$

and a prediction loss, $L_{pred}$, focused on Mean-Squared Error (MSE),

$$L_{pred} = \mathbb{E}_{t \sim B} \left\| \hat{z}_{t+1} - \bar{K}z_t - \bar{L}u_t \right\|^2. \tag{3}$$

These losses drive the learning of model parameters to accurately predict future latent states while learning representations from the visual input.

**b.) Koopman Operator and Eigenspetrum Initialization**: To construct representations that accurately capture the system dynamics, it is imperative to initialize the Koopman operator appropriately. The dynamic system's future state predictions can be expressed as:

$$\left[ \hat{z}_{(t+1)}^i, \ldots, \hat{z}_{(t+\tau)}^i \right] = \left[ \tilde{\lambda}^i, \ldots, \tilde{\lambda}_i^\tau \right] z_t^i + \left[ 1, \tilde{\lambda}^i, \ldots, \tilde{\lambda}_i^{\tau-1} \right] * \left[ c_t^i, \ldots, c_{t+\tau-1}^i \right] \tag{4}$$

where $\bar{\lambda}_i = \mu_i + i\omega_i$ denotes the eigenvalues of the diagonal Koopman operator. From a linear stability perspective, we would like to place the real values of these $\mu_i$ in a negative real plane with large values so that the system reaches asymptotic stability faster. However, this configuration may lead to exploding gradient issues, a direct consequence of the exponential increase in gradient magnitudes proportional to the eigenvalues' real parts.

**Lemma 1 (Exponential Scaling of Gradient Norms):** For the discrete dynamic system depicted in Equation 4, where $L_{pred} = \mathcal{L}_t$ is defined as the latent prediction loss in Equation 3, let $z_t$ represent the latent state and $c_t = \bar{L}u_t$ the control input at time step $t$. The gradient norms of $\mathcal{L}_t$ with respect to the $j$-th components of $z_{j,t}$ and $c_{j,t}$ exhibit exponential scaling compared to the gradient norms at time step $t+1$, governed by the real parts of the system's eigenvalues, $\mu_j$: $\left| \frac{\partial \mathcal{L}_t}{\partial z_{j,t}} \right| = e^{\Delta t \mu_j} \left| \frac{\partial \mathcal{L}_t}{\partial z_{j,t+1}} \right|$,

where $\Delta t$ denotes the time increment, and $\mu_j$ the real part of the $j$-th eigenvalue, underscoring the link between the gradient norms of the loss function, the latent representations, the control inputs, and the system's eigenstructure.

To mitigate potential stability issues and constrain gradient magnitudes, we limit the real parts of the eigenvalues, $\mu_j$, within the interval $[-0.5, -0.01]$, and initialize the imaginary parts, $\omega_j$, in ascending order of frequency as $\omega_j = j\pi$ similar to [17]. This approach ensures comprehensive frequency mode capture within the system dynamics, fostering stable and efficient learning of dynamic representations.

**c.) Task Conditioning with End-to-End learning** : To learn the contrastive spectral representations and spectral Koopman operator, we condition these networks to learn a linear effective controller

similar to [16]. Given Spectral Koopman embeddings $z = \psi_{\theta_k}(x)$ and its associated linear spectral latent system as shown in equation 4, we formulate a finite time horizon LQR problem in Koopman latent space as $\min_{u_{0:T}} \sum_{k=0}^{T} \left[ (z_k - z_{\text{ref}})^T Q(z_k - z_{\text{ref}}) + u_k^T R u_k \right]$ subject to $z_{k+1} = \bar{K} z_k + \bar{L} u_k$ (8) where $Q$ and $R$ are state and control cost diagonal matrices and $z_{\text{ref}}$ denotes the representation of goal input. We solve the above equation in an iterative procedure to recursively update the solution for a small number of iterations, typically $T < 10$, which is adequate to obtain a satisfactory and efficient approximation. The control policy is then given by $u \sim \pi_{\text{LQR}}(z|s) \triangleq \pi_{\text{LQR}}(z|\bar{K}, \bar{L}, Q, R)$ Further to optimize this controller towards the task, we draw inspiration from [9][16], we maximize the following objective via two Q value estimators as used in soft actor-critic method [15]: $\mathcal{L}_{\text{SAC}} = \mathbb{E}_{t \sim B} \left[ \min_{i=1,2} Q_i(z, u) - \alpha \log \pi_{\text{SAC}}(u|z) \right]$

For an end-to-end training of the network, we iteratively collect trajectory data batches from the task environment $E$ and use the three distinct loss objectives to train the networks in an end-to-end fashion parameter. The primary objective, $L_{\text{sac}}$ is used to optimize all the task parameters to learn efficient representations for the task. Additionally, we integrate contrastive learning loss $L_{\text{cst}}$ and model prediction loss $L_{\text{m}}$ to regularize the task parameter learning process and feature extraction.

# 4   Analytical Results

In this section, we aim to provide a theoretical analysis of the convergence behavior of our over network and show that our task network converges to an optimal policy given enough interactions with the environment. Specifically, we state the following theorems

**Theorem 1: (Convergence of Contrastive Loss via Gradient Descent)** *Let $\mathcal{L}_{cst}(\theta)$ be an L-smooth contrastive loss function for encoder parameters $\theta$ and assuming stochastic gradient descent (SGD) updates with learning rate $\alpha_t$ satisfying Robbins-Monro conditions. If $\hat{\nabla}_\theta \mathcal{L}_{cst}$ is an unbiased estimate of the gradient with bounded variance, then $\lim_{t \to \infty} \mathbb{E}[\|\nabla_\theta \mathcal{L}_{cst}(\theta_t)\|^2] = 0$. (Proof)*

**Theorem 2: (Convergence of Koopman Operator Approximations)**: *Given (i) a discrete-time linear dynamical system with states $\mathbf{z} \in \mathbb{R}^n$ and control inputs $\mathbf{u} \in \mathbb{R}^m$, evolving according to $\mathbf{z}_{k+1} = \mathbf{A}_{true}\mathbf{z}_k + \mathbf{B}_{true}\mathbf{u}_k$, where $\mathbf{A}_{true} \in \mathbb{R}^{n \times n}$ and $\mathbf{B}_{true} \in \mathbb{R}^{n \times m}$ are the true system matrices; and (ii) the Koopman operator approximation estimates $\mathbf{A}$ and $\mathbf{B}$ such that $\mathbf{z}_{k+1} \approx \mathbf{A}\mathbf{z}_k + \mathbf{B}\mathbf{u}_k$, the minimization of $\mathcal{L}_m(\mathbf{A}, \mathbf{B}; \mathbf{z}_k, \mathbf{u}_k, \mathbf{z}_{k+1})$ with respect to $\mathbf{A}$ and $\mathbf{B}$ converges to the true system matrices, i.e., $\lim_{n \to \infty} (\mathbf{A}, \mathbf{B}) = (\mathbf{A}_{true}, \mathbf{B}_{true})$, where $n$ represents the number of observations.*

**Theorem 3: (Convergence of the LQR Control Policy)** *Given a discrete-time linear system characterized by state transition matrix $\mathbf{A} \in \mathbb{R}^{n \times n}$ and control input matrix $\mathbf{B} \in \mathbb{R}^{n \times m}$ and the LQR problem aims to minimize a quadratic cost function $J = \sum_{k=0}^{\infty}(\mathbf{x}_k^\top \mathbf{Q}\mathbf{x}_k + \mathbf{u}_k^\top \mathbf{R}\mathbf{u}_k)$ with $\mathbf{Q} \geq 0$ and $\mathbf{R} > 0$, the DARE solution*

$$\mathbf{P}_{i+1} = \mathbf{A}^\top \mathbf{P}_i \mathbf{A} - \mathbf{A}^\top \mathbf{P}_i \mathbf{B} \left( \mathbf{R} + \mathbf{B}^\top \mathbf{P}_i \mathbf{B} \right)^{-1} \mathbf{B}^\top \mathbf{P}_i \mathbf{A} + \mathbf{Q},$$

*converges to $\mathbf{P}^*$, ensuring that the optimal control gains $\mathbf{G}^*$ yield a stable and optimal control policy.*

**Theorem 4 (LQR within SAC Optimizes Koopman Control Policy)**: *Let $\mathcal{L}_{sac}$ be the SAC loss for a given policy $\pi_{sac}(\mathbf{u}|\mathbf{z})$ integrated with the LQR control policy $\pi_{LQR}(\mathbf{z}|\mathbf{G})$ in a latent space $\mathbf{Z}$, derived via the Koopman operator theory for a nonlinear dynamical system. If the SAC loss $\mathcal{L}_{sac}$ is Lipschitz continuous with respect to the parameter set $\Omega = \{\mathbf{Q}, \mathbf{R}, \mathbf{A}, \mathbf{B}, \psi_\theta\}$ and $\mathcal{L}_{sac}$ is bounded below, then applying gradient descent updates on $\Omega$ to minimize $\mathcal{L}_{sac}$ guarantees convergence to a stationary point of $\mathcal{L}_{sac}$.*

Proofs for these theorems are provided in *Appendix Section C* Building on these theorems, it is established that our networks, given an adequate number of samples from the training data, are guaranteed to converge to an optimal task policy.

# 5 Empirical Results

In this section, we conduct simulated experiments primarily to explore these two questions: Firstly, Can our methodology effectively achieve desirable task performance using linear spectral Koopman representations across diverse environments exhibiting varying nonlinear dynamics? Secondly, Is it feasible to develop a globally linear model that aligns closely with the latent space, demonstrating resilience to noise and external disruptions, while maintaining efficiency in sampling and computation?

**Experimental Settings and Baselines** In our work, we implement the model that operates within a spectral Koopman space, specifically designed with a 128-dimensional complex space. This model incorporates a control embedding that maps control inputs to a 128-dimensional space as well, and all the models are trained for 100k and 500k timesteps. We use five distinct robotic control tasks from the DeepMind Control Suite [19], each characterized by varying state and action space dimensionalities: reacher easy, reacher hard, walker, ball in cup, cheetah, and cartpole. For comparative analysis, we implement several baseline models. CURL [9] is a model-free RL algorithm using a contrastive encoder for latent space control learning. Our approach, while also using a contrastive encoder, uniquely learns a complex-valued linear latent space. We also examine the To-KPM [16], which employs linear dense representations for dynamic modeling, whereas our model adopts a spectral Koopman form for efficient and stable system modeling. Additionally, we include model-based RL algorithms: TD-MPC [47] and PlaNet [30] use real-time planning with different variants of the Cross-Entropy Method (CEM), while Dreamer [29] employs background planning. SAC-State [15] serves as an upper performance bound, receiving direct state input from the simulator. Our experimental framework and the detailed analysis of these baselines are thoroughly documented in Appendix A.1 and A.2.

**Evaluation Metric**: In RL, agents learn from interactions with their environment, guided by task rewards. These rewards provide scalar feedback, allowing agents to refine policies for maximizing future rewards. For instance, in the cart pole system, the agent receives a fixed reward of +1 for every timestep the pole remains upright, within a predefined angle from the vertical. Therefore, the quicker the cartpole reaches a vertical position, the more rewards it will accumulate. In this work, we set the reward as negative of the LQR cost. The greater the reward, the lower the control cost, enabling a faster achievement of the desired state. We report the mean reward and the variance of the distribution across 5 random experiments for all the models.

Table 2: Performance comparison at 100K and 500K steps

| Model | 100K steps | | | | | 500K steps | | | | |
|---|---|---|---|---|---|---|---|---|---|---|
| | Reacher easy | Walker | Cartpole | Cheetah | Ball In Cup | Reacher easy | Walker | Cartpole | Cheetah | Ball In Cup |
| **Control in State Space - Upper bound performance** | | | | | | | | | | |
| Sac State | 919 ± 123 | 604 ± 317 | 812 ± 45 | 228 ± 95 | 957 ± 26 | 975 ± 5 | 964 ± 8 | 870 ± 7 | 772 ± 60 | 979 ± 6 |
| **Control in Pixel Space** | | | | | | | | | | |
| TDMPC [47] | 413 ± 62 | 653 ± 99 | 747 ± 78 | 274 ± 69 | 675 ± 221 | 722± 184 | 944 ± 15 | 860 ± 11 | 488 ± 74 | 967 ± 15 |
| Curl [9] | 460 ± 65 | 482 ± 28 | 547 ± 73 | 266 ± 27 | 741 ± 102 | 929 ± 44 | 897 ± 26 | 841 ± 45 | 518 ± 28 | 957 ± 6 |
| DrQ [48] | 601 ± 213 | 612 ± 164 | 759 ± 92 | **344 ± 67** | 913 ± 53 | 942 ± 71 | 921 ± 46 | 868 ± 10 | **660 ± 96** | 963 ± 9 |
| Dreamer [29] | 148 ± 53 | 216 ± 56 | 235 ± 73 | 159 ± 60 | 172 ± 96 | 581 ± 160 | 924 ± 35 | 711 ± 94 | 571 ± 109 | 964 ± 8 |
| Planet [30] | 140 ± 256 | 125 ± 57 | 303 ± 71 | 165 ± 123 | 198 ± 442 | 351 ± 483 | 293 ± 114 | 464 ± 50 | 321 ± 104 | 352 ± 467 |
| SLACv1 [49] | - | 361 ± 73 | - | 319 ± 56 | 512 ± 110 | - | 842 ± 51 | - | 640 ± 19 | 852 ± 71 |
| To-KPM [16] | 238 ± 352 | 414 ± 216 | 797 ± 35 | 14 ± 6 | 841 ± 167 | 968 ± 16 | 127 ± 59 | 835 ± 17 | 259 ± 18 | 921 ± 56 |
| SAC+AE [15] | 145 ± 30 | 42 ± 12 | 419 ± 40 | 197 ± 15 | 312 ± 63 | 145 ± 30 | 42 ± 12 | 419 ± 40 | 197 ± 15 | 312 ± 63 |
| Koopman AE [50] | 234 ± 11 | 320 ± 26 | 345 ± 10 | 261 ± 53 | 301 ± 47 | 327 ± 42 | 512 ± 68 | 400 ± 77 | 192 ± 31 | 412 ± 45 |
| Ours | **679 ± 300** | **640 ± 34** | **864 ± 4.2** | 305 ± 7.3 | **940 ± 36** | **969 ± 7.9** | **959 ± 15** | **874 ± 1.7** | 390 ± 6.9 | **967 ± 20** |

**Results** To operate in a sample-efficient regime, we trained our networks for only 100k steps, in contrast to the 500k-step training regimen utilized for all baseline models. Table 2 provides a detailed comparison of all the models for both 100k and 500k steps. For all the environments except Cheetah, our model achieves the highest reward, which notably corresponds to the negative of the control cost, outperforming all other models. We note that control models utilizing autoencoder-type prediction mechanisms, such as Koopman AE, SAC-AE, Dreamer and Planet, underperform, with SAC-AE, despite its similar RL exploration strategy to ours, performing particularly poorly. To-KPM, with

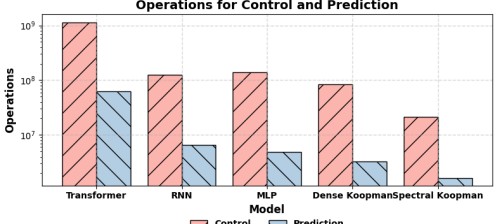
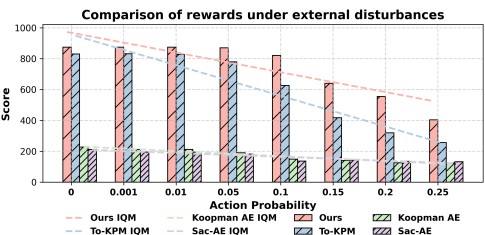

Figure 3: Computational Load of SOTA dynamical models

Figure 4: Performance of models with external disturbances

its dense Koopman model, also yields comparatively lower rewards than our model. Furthermore, when comparing our model against state-space model—specifically, state-based SAC —we observe that our approach surpasses the state-based controllers in terms of reward, even with a limited training duration of 100k steps, and exhibits very low trial variance for rewards. This underscores the high expressive power and task-guided representation capabilities of our model. In the cheetah task evaluation, however, our model has the third-best reward in the 100K step evaluation, only behind DrQ and SLACv1. This exception may be attributed to these models' nonlinear dynamics models with non-linear next step prediction and similar encoder to our model, making them optimized for state space exploration through deep RL. For more ablation studies of our model, please refer to Appendix D.

**Computational Efficiency for Linear vs Non-Linear dynamics** In this section, we evaluate the computational efficiency of state-of-the-art predictive models in conjunction with linearized control mechanisms. Specifically, we implement the MLP dynamics model as described in [9] and the Dense Koopman model from [16] for comparison. Additionally, to assess our method against parallelizable and causally structured models, we incorporate the Transformer model [51, 31] and a recurrent model outlined in [29], which are utilized to simulate system dynamics over more than 100 future steps while concurrently learning a linear control strategy for the system. Figure 3 illustrates the Multiply-Accumulate (MAC) operations required by each model, from which we deduce that our approach of using spectral Koopman necessitates the lowest computational effort for both control and prediction tasks. This comparison underscores the efficiency of our method in managing the computational demands associated with dynamic system modeling and control.

**Performance under external disturbances**: In this section, we introduce an external force to the cart pole system during evaluation, denoted by $F$, which follows a uniform distribution $F \sim$ Uniform($a_{\max}, 0, a_{\min}$). The probability of applying this force is represented by $p$, where $a_{\max}$ and $a_{\min}$ correspond to the environment's maximum and minimum allowable control input values, respectively. Figure 4 illustrates the mean reward and Interquartile Mean (IQM) for the models under study. Notably, our models demonstrate robust control capabilities, maintaining high performance even when the probability of external disturbances is increased to 0.25. This resilience is evidenced by a minimal degradation in performance when compared to alternative approaches.

## 6 Conclusion and Future Work

In this work, we introduce *RoboKoop*, a novel approach that integrates a contrastive encoder with spectral Koopman operators to learn visual representations guided by task learning. We demonstrate that *RoboKoop* surpasses current state-of-the-art methods, achieving superior performance while maintaining sample efficiency and robustness to noise and external disturbances. Looking forward, we aim to further enhance sample efficiency by exploring alternatives to Soft Actor-Critic (SAC) methods, potentially reducing the number of samples required for effective learning.

**Acknowledgments**

This work is based in part on research supported by SRC JUMP2.0 (CogniSense Center, Grant: 2023-JU-3133) and the Army Research Office (Grant: W911NF-19-1-0447). The views, opinions, findings, and conclusions expressed in this material are those of the author(s) and do not necessarily reflect the official policies or positions of SRC, the Army Research Office, or the U.S. Government.

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

## A  Model Details and Experimental Settings

### A.1  Simulation Environment for Empirical Study

In our research, we introduce *RoboKoop*, an algorithm distinguished by its sample efficiency, which processes pixel-based inputs to simultaneously learn linear dynamics and develop an effective control policy. This algorithm demonstrates versatility across a broad spectrum of environments. We have rigorously tested RoboKoop against continuous control challenges within the DeepMind Control Suite. Our selection of these particular tasks is grounded in several critical considerations:

1. Existing baseline methods exhibit suboptimal performance on these tasks, highlighting a gap that RoboKoop aims to fill.

2. Recent advancements have introduced both model-free and model-based strategies aimed at enhancing the sample efficiency of similar algorithms. Our work contributes to this ongoing dialogue by presenting an alternative approach.

3. The performance metrics obtained from these simulated tasks are highly indicative of real-world applicability, underscoring the practical relevance of our findings in broader contexts.

**Cartpole Swingup** This task is centered around the goal of swinging up a pole, initially in a downward orientation, attached to a moving cart, and then maintaining its upright position. Success in this task requires the precise application of forces to the cart, navigating through a 4D state space that represents the cart-pole system's kinematics, complemented by a 1D control space for force application.

**Cheetah Run** The objective here is to orchestrate the movements of a simulated planar cheetah to achieve rapid and stable running. This involves managing an 18D state space that encapsulates the kinematics of the cheetah's entire body, including its joints and limbs, while employing 6D torques as controls to manipulate the joints for optimal locomotion.

**Reacher** The Reacher task is designed to test precise motor control by requiring an agent to maneuver a simulated two-joint robotic arm to a target location in a 2D plane. This task involves navigating through an 11D state space that includes the positions and velocities of the arm's joints, as well as the position of the target. The control space is 2D, representing the torques applied at each joint. Success in this task is measured by the agent's ability to accurately and efficiently move the arm to the target position and maintain it there.

**Ball in Cup** In the Ball in Cup task, the objective is to control a simulated robot arm to swing and catch a ball attached to a string in a cup. This task is particularly challenging due to the nonlinear dynamics involved in swinging the ball and the precision required to catch it in the cup. The environment's state space is 8D, capturing the positions and velocities of the ball and the robot arm, as well as the angular position of the cup. The control space is 3D, representing the forces applied to the robot arm to achieve the desired swing motion. Success in this task requires a combination of dynamic coordination and precise timing.

**Walker** The Walker task involves controlling a bipedal robot to achieve stable and efficient locomotion. The state space for this task is 17D, encompassing the kinematic properties of the robot's body and legs, including joint positions and velocities. The control space is 6D, corresponding to the torques applied to the robot's joints. The objective is to navigate the robot through various terrains, maintaining balance and forward motion. Success in this task is determined by the robot's ability to move swiftly and stably without falling.

## A.2 Model Hyper parameters

Table 3 provides a comprehensive enumeration of the hyperparameters employed in our model, along with detailed descriptions of each parameter. For To-KPM [16] also, we use the same hyperparameters as our model for a fair evaluation.

## A.3 Algorithm Design

For an end-to-end training of the network, we iteratively collect trajectory data batches from the task environment $E$ and use the three distinct loss objectives to train the networks in an end-to-end fashion. The primary objective, $L_{\text{sac}}$, is used to optimize all the task parameters to learn efficient representations for the task. Additionally, we integrate contrastive learning loss $L_{\text{cst}}$ and model prediction loss $L_{\text{m}}$ to regularize the task parameter learning process and feature extraction. Algorithm 1 describes the loss computation and parameter updates for the model.

---

**Algorithm 1** Model Training

---

0: **Initialize** parameters: Koopman control $Q, R$, Dynamics $\bar{K}, \bar{L}$, Encoder $\psi_\theta$.
0: **Reset** task environment $E$.
0: **Initialize** data replay buffer $D$.
0: **for** each iteration $\eta$ **do**
0:   **Collect** roll-outs $\tau_\eta$ by executing policy $\pi_{\text{LQR}}$, store in $D$.
0:   **Sample** batch $B$ from $D$.
0:   **Compute** $\mathcal{L}_{\text{cst}}$: **Update** $\psi_\theta$ to minimize $\mathcal{L}_{\text{cst}}$.
0:   **Compute** $\mathcal{L}_{\text{sac}}$: **Update** $Q, R, \bar{K}, \bar{L}$ to minimize $\mathcal{L}_{\text{sac}}$ (excluding $\psi_\theta$).
0:   **Compute** $\mathcal{L}_{\text{m}}$: **Update** $\bar{K}, \bar{L}$ to minimize $\mathcal{L}_{\text{m}}$.
0: **end for**=0

---

Table 3: Hyperparameters and Configuration Details

| Name | Value | Description |
|------|-------|-------------|
| **Environment** | | |
| Pre transform image size | 100 | Initial size of images before applying transforms. |
| Frame stack | 3 | Number of frames stacked together as input. |
| Image size | 84 | The resolution of input images to the network. |
| Replay buffer capacity | 100000 | Maximum size of the replay buffer. |
| **Agent** | | |
| Hidden dim | 1024 | Dimension of hidden layers in neural networks. |
| Discount factor | 0.99 | Discount factor for future rewards ($\gamma$). |
| Init temperature | 0.1 | Initial temperature parameter for SAC algorithm. |
| Alpha lr | 0.0001 | Learning rate for the temperature parameter. |
| Alpha beta | 0.5 | Beta parameter for the Adam optimizer for temperature. |
| Actor lr | 0.001 | Learning rate for the actor network. |
| Actor beta | 0.9 | Beta parameter for the Adam optimizer for the actor network. |
| Actor update freq | 1 | Frequency of actor network updates. |
| Critic lr | 0.001 | Learning rate for the critic network. |
| Critic beta | 0.9 | Beta parameter for the Adam optimizer for the critic network. |
| Critic tau | 0.01 | Tau parameter for soft updates of the target networks. |
| Critic target update freq | 1 | Frequency of target network updates. |
| Encoder feature dim | 256 | Dimensionality of the encoded features. |
| Control encode dim | 128 | Dimensionality of the encoded control input. |
| Encoder lr | 0.001 | Learning rate for the encoder. |
| Encoder tau | 0.05 | Tau parameter for soft updates of the encoder. |
| Num layers | 4 | Number of layers in the convolutional neural networks. |
| Num filters | 32 | Number of filters in the first convolutional layer. |
| Curl latent dim | 128 | Dimensionality of the latent space in CURL. |
| Koopman update freq | 1 | Frequency of updating the Koopman operator. |
| Koopman fit optim lr | 0.001 | Learning rate for optimizing the Koopman operator. |
| Koopman fit coeff | 0.1 | Coefficient for fitting the Koopman operator. |
| Koopman horizon | 5 | Horizon length for Koopman predictions. |
| **Training** | | |
| Init steps | 1000 | Number of steps collected with random actions at the start of training. |
| Num train steps | 150000 | Total number of training steps. |
| Batch size | 128 | Batch size for training. |

# B   Baselines

This section delineates the comparative analysis of baselines utilized in our study and elucidates how our approach diverges from them.

## B.1   CURL: Contrastive Unsupervised Representations for Reinforcement Learning [9]

CURL, which stands for Contrastive Unsupervised Representations for Reinforcement Learning, employs contrastive learning to derive high-level features from raw pixels for reinforcement learning tasks. Our methodology, however, adopts a spectral Koopman operator model to explicitly learn system dynamics, a feature absent in CURL. This distinction permits an in-depth analysis of system stability and provides valuable insights into controller design. Unlike non-linear control policies that lack a comprehensive system analysis, linear systems can be thoroughly examined through eigenvalue analysis. We demonstrate this through a pole analysis of the Koopman operators in Section 5, highlighting the methodological differences and advantages.

## B.2   To-KPM [16]

To-KPM introduces a task-oriented approach that integrates a contrastive encoder with Koopman-based control. Unlike our model, To-KPM relies on a dense Koopman operator, leading to unstable poles and reduced sample efficiency due to the increased parameters required for learning the Koop-

man operator. These limitations are substantiated by the instability of poles (refer to Figures 4 and 5 in Section 5 of our paper) and underscore the efficiency of our approach.

### B.3 Planet [30]

Planet is a model-based agent that discerns environment dynamics directly from pixels, facilitating action selection through online planning within a compact latent space. The latent space is structured around a recurrent state-space model, which is computationally intensive, as evidenced in Section 5 (Figure 6). Additionally, its emphasis on multi-step prediction in pixel space compromises sample efficiency, necessitating extensive interactions with the environment.

### B.4 Koopman AE [20]

The Koopman AE methodology leverages a soft actor-critic policy, underpinned by a regularized autoencoder (AE), to learn a latent space model atop AE features. Unlike Planet, this approach also explicitly models dynamics using a Koopman operator. In contrast, our method eschews the use of VAEs or AEs for pixel reconstruction, opting instead to learn features via contrastive learning alone. This strategy ensures the prioritization of task-relevant features over the reconstruction of pixel space, enhancing task efficiency and model performance.

## C Analytical Results

### C.1 Convergence of Contrastive Learning

**Definitions and Assumptions**

1. **Smoothness:** The function $\mathcal{L}_{\mathrm{cst}}$ is assumed to be $L$-smooth with respect to $\theta$, meaning it has Lipschitz continuous gradients:

$$\|\nabla \mathcal{L}_{\mathrm{cst}}(\theta_1) - \nabla \mathcal{L}_{\mathrm{cst}}(\theta_2)\| \leq L\|\theta_1 - \theta_2\|, \quad \forall \theta_1, \theta_2.$$

2. **Unbiased Gradient Estimates:** The stochastic gradient $\hat{\nabla}_\theta \mathcal{L}_{\mathrm{cst}}$ is an unbiased estimate of the true gradient:

$$\mathbb{E}[\hat{\nabla}_\theta \mathcal{L}_{\mathrm{cst}}(\theta)] = \nabla_\theta \mathcal{L}_{\mathrm{cst}}(\theta).$$

3. **Bounded Variance:** The variance of the stochastic gradient is bounded by a constant $\sigma^2$:

$$\mathbb{E}[\|\hat{\nabla}_\theta \mathcal{L}_{\mathrm{cst}}(\theta) - \nabla_\theta \mathcal{L}_{\mathrm{cst}}(\theta)\|^2] \leq \sigma^2.$$

4. **Diminishing Learning Rates:** The learning rate $\alpha_t$ satisfies the Robbins-Monro conditions:

$$\sum_{t=1}^{\infty} \alpha_t = \infty, \quad \sum_{t=1}^{\infty} \alpha_t^2 < \infty.$$

**Convergence of Contrastive Loss via Gradient Descent**

**Theorem 1.:** *Let $\mathcal{L}_{cst}(\theta)$ be an $L$-smooth contrastive loss function for encoder parameters $\theta$ and assuming stochastic gradient descent (SGD) updates with learning rate $\alpha_t$ satisfying Robbins-Monro conditions. If $\hat{\nabla}_\theta \mathcal{L}_{cst}$ is an unbiased estimate of the gradient with bounded variance, then $\lim_{t \to \infty} \mathbb{E}[\|\nabla_\theta \mathcal{L}_{cst}(\theta_t)\|^2] = 0$.*

**Proof:** Given the Lipschitz continuity of $\psi_\theta$, and assuming the loss $\mathcal{L}_{\mathrm{cst}}$ inherits this property with respect to $\theta$, the Descent Lemma can be applied. The lemma states that for a Lipschitz continuous function $f$ with Lipschitz constant $L$,

$$f(x + \Delta x) \leq f(x) + \nabla f(x)^\top \Delta x + \frac{L}{2}\|\Delta x\|^2.$$

Given the $L$-smoothness of $\mathcal{L}_{\mathrm{cst}}$, we have for any $\theta_1, \theta_2$:

$$\mathcal{L}_{\text{cst}}(\theta_2) \leq \mathcal{L}_{\text{cst}}(\theta_1) + \nabla \mathcal{L}_{\text{cst}}(\theta_1)^\top (\theta_2 - \theta_1) + \frac{L}{2} \|\theta_2 - \theta_1\|^2.$$

Substituting the gradient descent update $\theta_{t+1} = \theta_t - \alpha_t \hat{\nabla}_\theta \mathcal{L}_{\text{cst}}(\theta_t)$:

$$\mathcal{L}_{\text{cst}}(\theta_{t+1}) \leq \mathcal{L}_{\text{cst}}(\theta_t) - \alpha_t \nabla \mathcal{L}_{\text{cst}}(\theta_t)^\top \hat{\nabla}_\theta \mathcal{L}_{\text{cst}}(\theta_t) + \frac{L\alpha_t^2}{2} \|\hat{\nabla}_\theta \mathcal{L}_{\text{cst}}(\theta_t)\|^2.$$

Taking expectations on both sides, and using the fact that $\mathbb{E}[\hat{\nabla}_\theta \mathcal{L}_{\text{cst}}] = \nabla_\theta \mathcal{L}_{\text{cst}}$ (unbiased gradient estimates) and the bounded variance assumption:

$$\mathbb{E}[\mathcal{L}_{\text{cst}}(\theta_{t+1})] \leq \mathbb{E}[\mathcal{L}_{\text{cst}}(\theta_t)] - \alpha_t \|\nabla_\theta \mathcal{L}_{\text{cst}}(\theta_t)\|^2 + \frac{L\alpha_t^2}{2}(\sigma^2 + \|\nabla_\theta \mathcal{L}_{\text{cst}}(\theta_t)\|^2).$$

Rearranging the terms, we aim to show that:

$$\alpha_t(1 - \frac{L\alpha_t}{2})\|\nabla_\theta \mathcal{L}_{\text{cst}}(\theta_t)\|^2 \leq \mathbb{E}[\mathcal{L}_{\text{cst}}(\theta_t)] - \mathbb{E}[\mathcal{L}_{\text{cst}}(\theta_{t+1})] + \frac{L\alpha_t^2 \sigma^2}{2}.$$

Given $\alpha_t$ satisfies the Robbins-Monro conditions and $1 - \frac{L\alpha_t}{2} > 0$ for sufficiently small $\alpha_t$, summing both sides over $t$ and applying the law of total expectation give:

$$\sum_{t=1}^\infty \alpha_t(1 - \frac{L\alpha_t}{2})\mathbb{E}[\|\nabla_\theta \mathcal{L}_{\text{cst}}(\theta_t)\|^2] \leq \mathcal{L}_{\text{cst}}(\theta_1) - \mathcal{L}_{\text{cst}}(\theta^*) + \sum_{t=1}^\infty \frac{L\alpha_t^2 \sigma^2}{2},$$

where $\theta^*$ is a local minimum of $\mathcal{L}_{\text{cst}}$.

Given the right-hand side is bounded (due to the boundedness of

$\mathcal{L}_{\text{cst}}$ and the conditions on $\alpha_t$), and $\sum_{t=1}^\infty \alpha_t(1 - \frac{L\alpha_t}{2}) = \infty$, it follows from the quasi-martingale convergence theorem and the Robbins-Monro conditions that:

$$\lim_{t \to \infty} \mathbb{E}[\|\nabla_\theta \mathcal{L}_{\text{cst}}(\theta_t)\|^2] = 0.$$

This implies that, in expectation, the gradient norm converges to 0, indicating convergence to a stationary point. Now using the Polyak-Łojasiewicz condition, it can be shown that this is a local minimum.

The exact form of $\mathcal{L}_{\text{cst}}$ and its gradient $\nabla_\theta \mathcal{L}_{\text{cst}}$. The Lipschitz constants for $\psi_\theta$ and $\mathcal{L}_{\text{cst}}$ Conditions under which the stochastic gradient is an unbiased estimate of the true gradient and has bounded variance. A suitable learning rate schedule $\alpha_t$ that guarantees convergence.

### C.2 Stability and Convergence of the Koopman Operator Approximation

**Theorem 2: Convergence of Koopman Operator Approximations**: Given (i) a discrete-time linear dynamical system with states $\mathbf{z} \in \mathbb{R}^n$ and control inputs $\mathbf{u} \in \mathbb{R}^m$, evolving according to $\mathbf{z}_{k+1} = \mathbf{A}_{true}\mathbf{z}_k + \mathbf{B}_{true}\mathbf{u}_k$, where $\mathbf{A}_{true} \in \mathbb{R}^{n \times n}$ and $\mathbf{B}_{true} \in \mathbb{R}^{n \times m}$ are the true system matrices; and (ii) the Koopman operator approximation approach, which seeks to estimate matrices $\mathbf{A}$ and $\mathbf{B}$ such that $\mathbf{z}_{k+1} \approx \mathbf{A}\mathbf{z}_k + \mathbf{B}\mathbf{u}_k$, based on a loss function $\mathcal{L}_m(\mathbf{A}, \mathbf{B}; \mathbf{z}_k, \mathbf{u}_k, \mathbf{z}_{k+1})$, the minimization of $\mathcal{L}_m$ with respect to $\mathbf{A}$ and $\mathbf{B}$ over the observed data converges to the true system matrices, i.e.,

$$\lim_{n \to \infty} (\mathbf{A}, \mathbf{B}) = (\mathbf{A}_{true}, \mathbf{B}_{true}),$$

where $n$ represents the number of observations.

**Proof**

We model the evolution of the system's state as a linear regression problem, where:

- $\mathbf{Z}_{\text{next}}$ is the matrix of next states $\mathbf{z}_{k+1}$,
- $\mathbf{X}$ is the design matrix composed of current states $\mathbf{z}_k$ and control inputs $\mathbf{u}_k$,
- $\boldsymbol{\Theta}$ is the parameters matrix to be estimated, combining $\mathbf{A}$ and $\mathbf{B}$,
- $\epsilon$ is the error term.

The equation $\mathbf{Z}_{\text{next}} = \mathbf{X}\boldsymbol{\Theta} + \epsilon$ encapsulates this linear relationship.

The objective function to minimize the difference between the predicted next states and the actual next states, quantified by the Frobenius norm of their difference can be written as:

$$\mathcal{L}_m = \|\mathbf{Z}_{\text{next}} - \mathbf{X}\boldsymbol{\Theta}\|_F^2,$$

where $\|\cdot\|_F$ denotes the Frobenius norm. To minimize $\mathcal{L}_m$, we calculate the gradient of the loss function with respect to $\boldsymbol{\Theta}$ and set it to zero: $\nabla_{\boldsymbol{\Theta}}\mathcal{L}_m = -2\mathbf{X}^\top(\mathbf{Z}_{\text{next}} - \mathbf{X}\boldsymbol{\Theta}) = 0$.

Solving this equation for $\boldsymbol{\Theta}$ gives: $\boldsymbol{\Theta} = (\mathbf{X}^\top\mathbf{X})^{-1}\mathbf{X}^\top\mathbf{Z}_{\text{next}}$. This is the least squares solution, providing the best estimate of $\boldsymbol{\Theta}$ given the data.

With the assumption that the observations $\mathbf{X}$ and $\mathbf{Z}_{\text{next}}$ sufficiently cover the entire state and control input space and as the number of observations $n$ approaches infinity ($N \to \infty$), the matrices $\mathbf{X}^\top\mathbf{X}$ and $\mathbf{X}^\top\mathbf{Z}_{\text{next}}$ will converge to their expected values. This ensures that the estimated parameters $\boldsymbol{\Theta}$, which combine $\mathbf{A}$ and $\mathbf{B}$, converge to the true system matrices $\mathbf{A}_{true}$ and $\mathbf{B}_{true}$ that govern the system's dynamics.

The solution involves setting the gradient of $\mathcal{L}_m$ with respect to $\boldsymbol{\Theta}$ to zero, leading to:

$$\nabla_{\boldsymbol{\Theta}}\mathcal{L}_m = -2\mathbf{X}^\top(\mathbf{Z}_{\text{next}} - \mathbf{X}\boldsymbol{\Theta}) = 0$$

Solving this equation yields the estimate for $\boldsymbol{\Theta}$:

$$\boldsymbol{\Theta} = (\mathbf{X}^\top\mathbf{X})^{-1}\mathbf{X}^\top\mathbf{Z}_{\text{next}}$$

Given a sufficiently diverse and large dataset ($n \to \infty$), the estimates converge to the true system dynamics because the matrices $\mathbf{X}^\top\mathbf{X}$ and $\mathbf{X}^\top\mathbf{Z}_{\text{next}}$ approach their expected values, ensuring the estimated parameters ($\mathbf{A}$ and $\mathbf{B}$) converge to the true parameters ($\mathbf{A}_{true}$ and $\mathbf{B}_{true}$).

This proof assumes sufficient data coverage across the state and control input space, which guarantees the convergence of the Koopman operator approximations to the true system dynamics, thereby validating the theorem.

### C.3 Convergence of the LQR Control Policy

**Theorem 3: Convergence of the LQR Control Policy** Given a discrete-time linear system characterized by state transition matrix $\mathbf{A} \in \mathbb{R}^{n \times n}$ and control input matrix $\mathbf{B} \in \mathbb{R}^{n \times m}$ and the LQR problem aims to minimize a quadratic cost function $J = \sum_{k=0}^{\infty}(\mathbf{x}_k^\top\mathbf{Q}\mathbf{x}_k + \mathbf{u}_k^\top\mathbf{R}\mathbf{u}_k)$ with $\mathbf{Q} \geq 0$ and $\mathbf{R} > 0$, the iterative solution to the Discrete-time Algebraic Riccati Equation (DARE)

$$\mathbf{P}_{i+1} = \mathbf{A}^\top\mathbf{P}_i\mathbf{A} - \mathbf{A}^\top\mathbf{P}_i\mathbf{B}\left(\mathbf{R} + \mathbf{B}^\top\mathbf{P}_i\mathbf{B}\right)^{-1}\mathbf{B}^\top\mathbf{P}_i\mathbf{A} + \mathbf{Q},$$

converges to the optimal solution $\mathbf{P}^*$ for the LQR problem, ensuring that the optimal control gains $\mathbf{G}^* = -(\mathbf{R} + \mathbf{B}^\top\mathbf{P}^*\mathbf{B})^{-1}\mathbf{B}^\top\mathbf{P}^*\mathbf{A}$ yield a stable and optimal control policy.

**Proof:**

To prove the convergence of the Linear Quadratic Regulator (LQR) control policy, we focus on the discrete-time setting, where the goal is to design an optimal control policy that minimizes a given cost function. The essence of the proof involves showing that the solution to the Discrete-time Algebraic Riccati Equation (DARE) converges to a unique positive semidefinite matrix, which then defines the optimal control gains.

We are given a discrete-time linear system:

$$\mathbf{x}_{k+1} = \mathbf{A}\mathbf{x}_k + \mathbf{B}\mathbf{u}_k,$$

and aim to minimize the infinite-horizon quadratic cost function:

$$J = \sum_{k=0}^{\infty} \left( \mathbf{x}_k^\top \mathbf{Q}\mathbf{x}_k + \mathbf{u}_k^\top \mathbf{R}\mathbf{u}_k \right),$$

where $\mathbf{Q} \geq 0$ (positive semidefinite) and $\mathbf{R} > 0$ (positive definite) are the state and control weight matrices, respectively.

The optimal control policy for this problem can be derived using dynamic programming, leading to the DARE:

$$\mathbf{P} = \mathbf{A}^\top \mathbf{P}\mathbf{A} - \mathbf{A}^\top \mathbf{P}\mathbf{B} \left( \mathbf{R} + \mathbf{B}^\top \mathbf{P}\mathbf{B} \right)^{-1} \mathbf{B}^\top \mathbf{P}\mathbf{A} + \mathbf{Q},$$

where $\mathbf{P}$ is the solution that defines the optimal cost-to-go matrix.

The convergence of the LQR control policy essentially means proving that the iterative solution to the DARE converges to a unique positive semidefinite matrix $\mathbf{P}^*$. Here are the key steps:

1. **Monotonicity and Boundedness:**

To prove that the sequence $\{\mathbf{P}_i\}$ generated by the Discrete-time Algebraic Riccati Equation (DARE) iterations is monotonically decreasing and bounded below, thus ensuring convergence, let's delve into equations and inequalities that illustrate these properties. Consider the iterative update rule for the DARE:

$$\mathbf{P}_{i+1} = \mathbf{A}^\top \mathbf{P}_i\mathbf{A} - \mathbf{A}^\top \mathbf{P}_i\mathbf{B} \left( \mathbf{R} + \mathbf{B}^\top \mathbf{P}_i\mathbf{B} \right)^{-1} \mathbf{B}^\top \mathbf{P}_i\mathbf{A} + \mathbf{Q},$$

where:

- $\mathbf{A}$ and $\mathbf{B}$ define the system dynamics, - $\mathbf{R}$ is the control weighting matrix, which is positive definite ($\mathbf{R} > 0$), - $\mathbf{Q}$ is the state weighting matrix, which is positive semidefinite ($\mathbf{Q} \geq 0$), - $\mathbf{P}_i$ is the cost-to-go matrix at iteration $i$.

To show that $\mathbf{P}_{i+1} \leq \mathbf{P}_i$, we need to establish that $\mathbf{P}_i - \mathbf{P}_{i+1}$ is positive semidefinite for each $i$. The Riccati update aims to minimize the cost function $J_i$ associated with using the control law derived from $\mathbf{P}_i$. Therefore, if we define the cost reduction as $\Delta\mathbf{P}_i = \mathbf{P}_i - \mathbf{P}_{i+1}$, we seek to show that $\Delta\mathbf{P}_i \geq 0$ (i.e., $\Delta\mathbf{P}_i$ is positive semidefinite).

Starting from the DARE update rule and rearranging terms gives us:

$$\Delta\mathbf{P}_i = \mathbf{P}_i - \mathbf{P}_{i+1} = \mathbf{A}^\top \mathbf{P}_i\mathbf{B} \left( \mathbf{R} + \mathbf{B}^\top \mathbf{P}_i\mathbf{B} \right)^{-1} \mathbf{B}^\top \mathbf{P}_i\mathbf{A},$$

Given that $\mathbf{R} > 0$ and $\mathbf{P}_i$ is positive semidefinite, it follows that the right-hand side of the equation above is positive semidefinite. This is because the term inside the parenthesis, $\mathbf{R} + \mathbf{B}^\top \mathbf{P}_i\mathbf{B}$, is positive definite, making its inverse also positive definite, and thus $\Delta\mathbf{P}_i$ is positive semidefinite, indicating that $\mathbf{P}_{i+1} \leq \mathbf{P}_i$.

The sequence is bounded below by the zero matrix, given that the cost-to-go matrices $\mathbf{P}_i$ represent quadratic cost functions which are non-negative:

$$\mathbf{P}_i \geq 0 \quad \forall i,$$

implying that the sequence cannot decrease indefinitely and is bounded below by a matrix where all elements are greater than or equal to zero. Given the monotonicity and boundedness of the sequence $\{\mathbf{P}_i\}$, it follows from the Monotone Convergence Theorem for matrices that the sequence converges to a limit, say $\mathbf{P}^*$, which is the solution to the DARE and represents the optimal cost-to-go matrix:

$$\lim_{i \to \infty} \mathbf{P}_i = \mathbf{P}^*,$$

where $\mathbf{P}^*$ satisfies the DARE and thus confirms the optimality and stability of the LQR control policy derived from it.

By establishing the monotonic decrease and boundedness below of the sequence $\{\mathbf{P}_i\}$, we have shown that this sequence converges to a matrix $\mathbf{P}^*$ that minimizes the LQR cost function. This $\mathbf{P}^*$ is the fixed point of the DARE, providing the optimal cost-to-go estimate and ensuring the stability and optimality of the LQR control policy derived from it.

2. **Fixed Point Convergence:** Under the assumptions that $\mathbf{A}$, $\mathbf{B}$, $\mathbf{Q}$, and $\mathbf{R}$ satisfy certain controllability and observability conditions, it can be shown that the iteration converges to a fixed point. To prove that the limit of the sequence $\{\mathbf{P}_i\}$, denoted as $\mathbf{P}^*$, satisfies the Discrete-time Algebraic Riccati Equation (DARE) and is thus a fixed point of the iteration process, we employ the properties of convergence and continuity of matrix operations.

Given the iterative process:

$$\mathbf{P}_{i+1} = \mathbf{A}^\top \mathbf{P}_i \mathbf{A} - \mathbf{A}^\top \mathbf{P}_i \mathbf{B} \left(\mathbf{R} + \mathbf{B}^\top \mathbf{P}_i \mathbf{B}\right)^{-1} \mathbf{B}^\top \mathbf{P}_i \mathbf{A} + \mathbf{Q},$$

we aim to show that, as $i \to \infty$, $\mathbf{P}_i \to \mathbf{P}^*$ and that $\mathbf{P}^*$ satisfies the DARE:

$$\mathbf{P}^* = \mathbf{A}^\top \mathbf{P}^* \mathbf{A} - \mathbf{A}^\top \mathbf{P}^* \mathbf{B} \left(\mathbf{R} + \mathbf{B}^\top \mathbf{P}^* \mathbf{B}\right)^{-1} \mathbf{B}^\top \mathbf{P}^* \mathbf{A} + \mathbf{Q}.$$

From previous steps, we have shown that the sequence $\{\mathbf{P}_i\}$ is monotonically decreasing and bounded below, which guarantees convergence to a limit $\mathbf{P}^*$ due to the Monotone Convergence Theorem for matrices.

The operations involved in the iterative update rule, including matrix addition, multiplication, and inversion, are continuous functions of their arguments. This means that if a sequence of matrices $\{\mathbf{X}_i\}$ converges to $\mathbf{X}$, then the limit of a continuous function $f(\mathbf{X}_i)$ is $f(\mathbf{X})$. The update rule can be seen as the application of a continuous function $f$ to $\mathbf{P}_i$:

$$f(\mathbf{P}_i) = \mathbf{A}^\top \mathbf{P}_i \mathbf{A} - \mathbf{A}^\top \mathbf{P}_i \mathbf{B} \left(\mathbf{R} + \mathbf{B}^\top \mathbf{P}_i \mathbf{B}\right)^{-1} \mathbf{B}^\top \mathbf{P}_i \mathbf{A} + \mathbf{Q}.$$

Given the convergence $\mathbf{P}_i \to \mathbf{P}^*$, by continuity, we have:

$$\lim_{i \to \infty} f(\mathbf{P}_i) = f(\lim_{i \to \infty} \mathbf{P}_i) = f(\mathbf{P}^*).$$

This implies:

$$\mathbf{P}^* = \mathbf{A}^\top \mathbf{P}^* \mathbf{A} - \mathbf{A}^\top \mathbf{P}^* \mathbf{B} \left(\mathbf{R} + \mathbf{B}^\top \mathbf{P}^* \mathbf{B}\right)^{-1} \mathbf{B}^\top \mathbf{P}^* \mathbf{A} + \mathbf{Q},$$

which is precisely the DARE. By showing that $\mathbf{P}^*$ satisfies the DARE, we've proven that $\mathbf{P}^*$ is a fixed point of the iteration process. This fixed point represents the solution to the DARE, establishing the optimality of the limit matrix $\mathbf{P}^*$ for the LQR problem.

Thus, by leveraging the properties of monotonicity, boundedness, convergence, and the continuity of matrix operations, we've demonstrated that the limit of the sequence $\{\mathbf{P}_i\}$, $\mathbf{P}^*$, satisfies the Discrete-time Algebraic Riccati Equation, making it the optimal solution and a fixed point of the iterative process.

The convergence of the LQR control policy to an optimal solution involves demonstrating that the iterative solution to the DARE converges to a unique matrix that minimizes the cost function and that the corresponding control policy stabilizes the system. The proof relies on algebraic properties of the Riccati equation, control theory, and the system's controllability and observability conditions.

### C.4 Integration of LQR within SAC Framework Optimizes Koopman Control Policy

**Lemma:** Given a loss function $\mathcal{L}$ that is Lipschitz continuous with respect to the parameters $\mathbf{\Omega}$, and bounded below, the sequence $\{\mathbf{\Omega}_t\}$ generated by the gradient descent updates:

$$\mathbf{\Omega}_{t+1} = \mathbf{\Omega}_t - \eta \nabla_{\mathbf{\Omega}} \mathcal{L}(\mathbf{\Omega}_t),$$

with a sufficiently small, fixed learning rate $\eta > 0$, converges to a stationary point $\mathbf{\Omega}^*$, where $\nabla_{\mathbf{\Omega}} \mathcal{L}(\mathbf{\Omega}^*) = 0$.

**Proof:** Given that $\mathcal{L}$ is Lipschitz continuous with Lipschitz constant $L$, we have for the gradient descent update:

$$\mathcal{L}(\mathbf{\Omega}_{t+1}) \leq \mathcal{L}(\mathbf{\Omega}_t) + \nabla_{\mathbf{\Omega}} \mathcal{L}(\mathbf{\Omega}_t)^\top (\mathbf{\Omega}_{t+1} - \mathbf{\Omega}_t) + \frac{L}{2} \|\mathbf{\Omega}_{t+1} - \mathbf{\Omega}_t\|^2. \tag{5}$$

$$\Rightarrow \mathbf{\Omega}_{t+1} - \mathbf{\Omega}_t = -\eta \nabla_{\mathbf{\Omega}} \mathcal{L}(\mathbf{\Omega}_t). \tag{6}$$

$$\Rightarrow \mathcal{L}(\mathbf{\Omega}_{t+1}) \leq \mathcal{L}(\mathbf{\Omega}_t) - \eta \|\nabla_{\mathbf{\Omega}} \mathcal{L}(\mathbf{\Omega}_t)\|^2 + \frac{L\eta^2}{2} \|\nabla_{\mathbf{\Omega}} \mathcal{L}(\mathbf{\Omega}_t)\|^2. \tag{7}$$

Choosing $\eta$: Select $\eta$ such that $0 < \eta < \frac{2}{L}$, ensuring that:

$$\mathcal{L}(\mathbf{\Omega}_{t+1}) \leq \mathcal{L}(\mathbf{\Omega}_t) - \left(\eta - \frac{L\eta^2}{2}\right) \|\nabla_{\mathbf{\Omega}} \mathcal{L}(\mathbf{\Omega}_t)\|^2.$$

Since $\mathcal{L}$ is bounded below, and $\mathcal{L}(\mathbf{\Omega}_{t+1}) \leq \mathcal{L}(\mathbf{\Omega}_t)$ for all $t$, the sequence $\{\mathcal{L}(\mathbf{\Omega}_t)\}$ is non-increasing and bounded. This implies convergence of the loss function values.

The reduction of the loss at each step is proportional to the square of the norm of the gradient. If the sequence $\{\mathbf{\Omega}_t\}$ did not converge to a stationary point, the gradient norm would not approach zero, contradicting the boundedness and convergence of the loss function values. Therefore, the gradient norm must approach zero, i.e., $\nabla_{\mathbf{\Omega}} \mathcal{L}(\mathbf{\Omega}^*) = 0$, indicating convergence to a stationary point.

**Theorem 4**: *Let $\mathcal{L}_{sac}$ be the Soft Actor-Critic (SAC) loss function for a given policy $\pi_{sac}(\mathbf{u}|\mathbf{z})$ integrated with the Linear Quadratic Regulator (LQR) control policy $\pi_{LQR}(\mathbf{z}|\mathbf{G})$ in a latent space $\mathbf{Z}$, derived via the Koopman operator theory for a nonlinear dynamical system. If the SAC loss $\mathcal{L}_{sac}$ is Lipschitz continuous with respect to the parameter set $\mathbf{\Omega} = \{\mathbf{Q}, \mathbf{R}, \mathbf{A}, \mathbf{B}, \psi_\theta\}$ and $\mathcal{L}_{sac}$ is bounded below, then applying gradient descent updates on $\mathbf{\Omega}$ to minimize $\mathcal{L}_{sac}$ guarantees convergence to a stationary point of $\mathcal{L}_{sac}$.*

**Proof:** Assume $\mathcal{L}_{\text{sac}}$ satisfies the Lipschitz condition with Lipschitz constant $L > 0$, i.e.,

$$|\mathcal{L}_{\text{sac}}(\mathbf{\Omega}_1) - \mathcal{L}_{\text{sac}}(\mathbf{\Omega}_2)| \leq L \|\mathbf{\Omega}_1 - \mathbf{\Omega}_2\|,$$

for any $\boldsymbol{\Omega}_1, \boldsymbol{\Omega}_2$ in the parameter space.

Now, the update rule for the parameters $\boldsymbol{\Omega}$ via gradient descent is given by:

$$\boldsymbol{\Omega}_{t+1} = \boldsymbol{\Omega}_t - \eta \nabla_{\boldsymbol{\Omega}} \mathcal{L}_{\mathrm{sac}}(\boldsymbol{\Omega}_t),$$

where $\eta > 0$ is the learning rate.

Using Lemma 1, given $\mathcal{L}_{\mathrm{sac}}$ is bounded below and Lipschitz continuous, the sequence $\{\boldsymbol{\Omega}_t\}$ produced by the gradient descent updates will converge to a stationary point $\boldsymbol{\Omega}^*$, characterized by:

$$\nabla_{\boldsymbol{\Omega}} \mathcal{L}_{\mathrm{sac}}(\boldsymbol{\Omega}^*) = 0.$$

Hence we show the optimality and stability via LQR Integration. The integration of the LQR policy $\pi_{\mathrm{LQR}}$ ensures that within the linear approximation of the dynamical system dynamics in the latent space $\mathbf{Z}$, the SAC framework, enhanced with LQR, converges towards optimal control actions. The LQR component provides an optimal control policy for linearized dynamics around the current state and control, ensuring that the SAC algorithm's policy updates enhance both stability and optimality in control decisions.

For a linear system $\mathbf{z}_{k+1} = \mathbf{A}\mathbf{z}_k + \mathbf{B}\mathbf{u}_k$, the LQR aims to minimize the cost function:

$$J = \sum_{k=0}^{\infty} \left( \mathbf{z}_k^\top \mathbf{Q}\mathbf{z}_k + \mathbf{u}_k^\top \mathbf{R}\mathbf{u}_k \right),$$

where $\mathbf{Q} \geq 0$ and $\mathbf{R} > 0$. The optimal control law is $\mathbf{u}_k^* = -\mathbf{K}\mathbf{z}_k$ with $\mathbf{K} = (\mathbf{R} + \mathbf{B}^\top \mathbf{P}\mathbf{B})^{-1}\mathbf{B}^\top \mathbf{P}\mathbf{A}$, where $\mathbf{P}$ solves the Algebraic Riccati Equation (ARE):

$$\mathbf{P} = \mathbf{A}^\top \mathbf{P}\mathbf{A} - \mathbf{A}^\top \mathbf{P}\mathbf{B}(\mathbf{R} + \mathbf{B}^\top \mathbf{P}\mathbf{B})^{-1}\mathbf{B}^\top \mathbf{P}\mathbf{A} + \mathbf{Q}.$$

The SAC algorithm seeks to optimize the policy $\pi_{\mathrm{sac}}(\mathbf{u}|\mathbf{z})$ by solving:

$$\max_{\pi} \mathbb{E} \left[ \sum_{k=0}^{\infty} \gamma^k \left( R(\mathbf{z}_k, \mathbf{u}_k) + \alpha \mathcal{H}(\pi(\cdot|\mathbf{z}_k)) \right) \right],$$

where $\mathcal{H}$ denotes the entropy of the policy, promoting exploration, and $\alpha$ is the temperature parameter that balances reward and entropy.

Integration means adjusting the SAC optimization to include the LQR solution as a baseline or regularization term. The objective becomes:

$$\max_{\pi} \mathbb{E} \left[ \sum_{k=0}^{\infty} \gamma^k \left( R(\mathbf{z}_k, \mathbf{u}_k) + \alpha \mathcal{H}(\pi(\cdot|\mathbf{z}_k)) - \lambda J_{\mathrm{LQR}}(\mathbf{z}_k, \mathbf{u}_k) \right) \right],$$

where $\lambda$ is a weighting coefficient, and $J_{\mathrm{LQR}}$ is the LQR cost function introduced above. This formulation explicitly guides the SAC policy towards the LQR's optimal policy within the linear approximation of the dynamics.

The optimal policy $\pi^*$ and the corresponding control law $\mathbf{u}^*$ from this integrated optimization problem are given by (1) the policy $\pi^*$ that maximizes the augmented objective, and (2) the control law that minimizes the LQR cost, ensuring stability as $\mathbf{P}$ guarantees the eigenvalues of $(\mathbf{A} - \mathbf{B}\mathbf{K})$ lie within the unit circle, ensuring the system's stability.

The parameter update rule incorporating both SAC optimization and LQR regularization is given by:

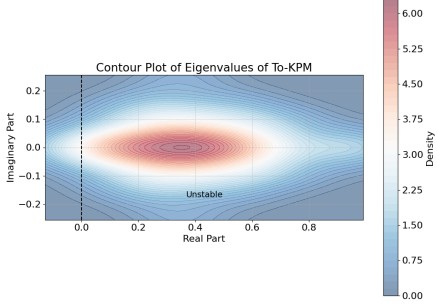

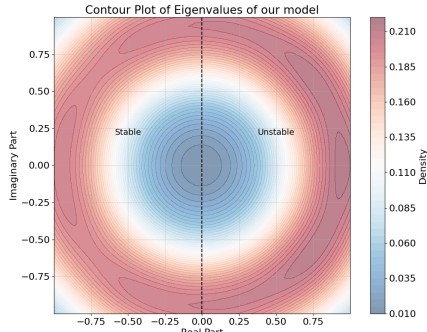

Figure 5: Eigenspectrum of To-KPM        Figure 6: Eigenspectrum of our model

$$\boldsymbol{\Omega}_{t+1} = \boldsymbol{\Omega}_t - \eta \nabla_{\boldsymbol{\Omega}} \left( \mathcal{L}_{\mathrm{sac}}(\boldsymbol{\Omega}_t) - \lambda J_{\mathrm{LQR}}(\boldsymbol{\Omega}_t) \right),$$

where $\mathcal{L}_{\mathrm{sac}}$ and $J_{\mathrm{LQR}}$ are differentiable with respect to $\boldsymbol{\Omega}$, ensuring that the gradient descent steps move the parameters towards minimizing the SAC loss while adhering to the LQR optimality criteria. Given the Lipschitz continuity and differentiability of $\mathcal{L}_{\mathrm{sac}} - \lambda J_{\mathrm{LQR}}$, the updates guarantee convergence to a stationary point $\boldsymbol{\Omega}^*$ where $\nabla_{\boldsymbol{\Omega}} \left( \mathcal{L}_{\mathrm{sac}}(\boldsymbol{\Omega}^*) - \lambda J_{\mathrm{LQR}}(\boldsymbol{\Omega}^*) \right) = 0$, encapsulating both the optimal policy in the SAC framework and the stability provided by the LQR control law.

Thus, we've shown how this combined approach integrating LQR within the SAC framework leverages LQR's optimality and stability, guiding the policy updates in SAC towards enhanced control decisions. The integration explicitly incorporates the LQR's linear control optimality into SAC's nonlinear policy optimization, ensuring convergence towards optimal and stable control actions in the latent space $\mathbf{Z}$. Thus, we show that under the conditions of Lipschitz continuity and boundedness of the SAC loss function, gradient descent optimization of the combined SAC and LQR policies in the Koopman latent space converges to a stationary point, optimizing the overall Koopman control policy. This integration not only leverages the strengths of both SAC and LQR but also ensures that the optimization process is theoretically grounded and guaranteed to reach a point of stability and optimality.

# D    Empirical Results

In this section, we conduct an ablation study to identify which components of our network contribute to its superior performance with a limited number of training steps. First, to demonstrate the effect of nonlinearity, we use CURL[9] as a baseline. CURL features a contractive encoder similar to ours but employs nonlinear dynamics, unlike our spectral dynamics. For comparison with a linear dense model, we use ToKPM[16], which relies on dense linear dynamics as opposed to our spectral model. Throughout the ablation studies, we demonstrate that our model outperforms both baselines. For this section, we present the results for models trained for 150,000 steps, as the other baselines showed poor performance when evaluated at 100,000 time steps.

## D.1    Eigenspectrum of our model

In Figures 5 and 6, we present the eigenspectrum contour plots for the To-KPM model and our proposed model, respectively. Analysis reveals that the eigenvalues of the To-KPM model predominantly reside on the positive real axis, with an average value of approximately 0.4. Conversely, our model exhibits a symmetric distribution of eigenvalues across the imaginary axis, featuring an equal proportion of positive and negative real eigenvalues. This distribution aligns with an increasing trend of eigenvalues as per $\omega = j\pi$. Within the framework of the Koopman operator theory,

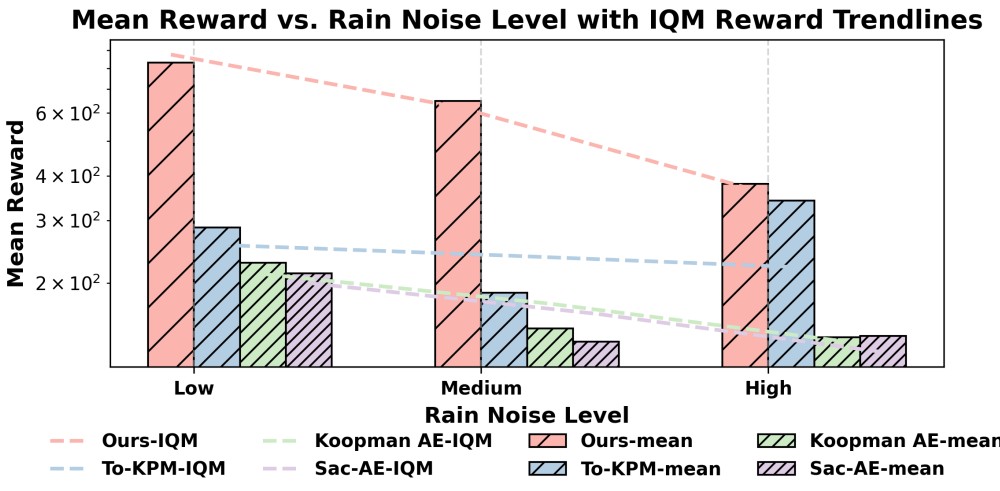

Figure 7: Performance of models in rainy conditions.

negative eigenvalues signify that the system's observables exhibit exponential decay over time, as these eigenvalues are integral to the exponential term in the solution to the linear system governed by the Koopman operator. Hence, negative eigenvalues are indicative of stable observable behaviors, whereas positive eigenvalues suggest exponential growth in observables, pointing to instability. The presence of positive eigenvalues in the To-KPM model undermines its ability to learn stable representations from images using a finite-dimensional Koopman operator, leading to inferior performance compared to our model, which benefits from a balanced distribution of positive and negative eigenvalues.

### D.2 Ablation Study on Spectral Koopman Operator Initialization

In this section, we conduct an ablation study to evaluate various initialization strategies for the Koopman spectral method, as detailed in Section 3.2. Our baseline configuration sets the Koopman operator's real value at -0.2, with frequencies arranged in increasing order. To assess the impact of initialization on performance, we explore three additional designs: (a) learnable real values with increasing frequency, (b) constant real values with random frequency, and (c) learnable real values with random frequency. This examination seeks to identify the initialization method that most effectively enhances the accuracy and stability of the spectral Koopman method.

Table 4 presents the mean reward for all the models across cheetah and cartpole simulations. Our analysis reveals that the strategy of employing constant real values with increasing frequency for the imaginary component of the initialization yields superior results.

Table 4: Summary of Experimental Results for Different Model Initializations

| Model Initialization | | Cartpole | Cheetah |
|---|---|---|---|
| $\mu_i$ | $\omega_i$ | Reward | Reward |
| Constant | Random | 85 | 21.36 |
| Learnable | Random | 85 | 9.04 |
| Learnable | Increasing freq | 155 | 285 |
| **Constant** | **Increasing freq** | **874** | **311.19** |

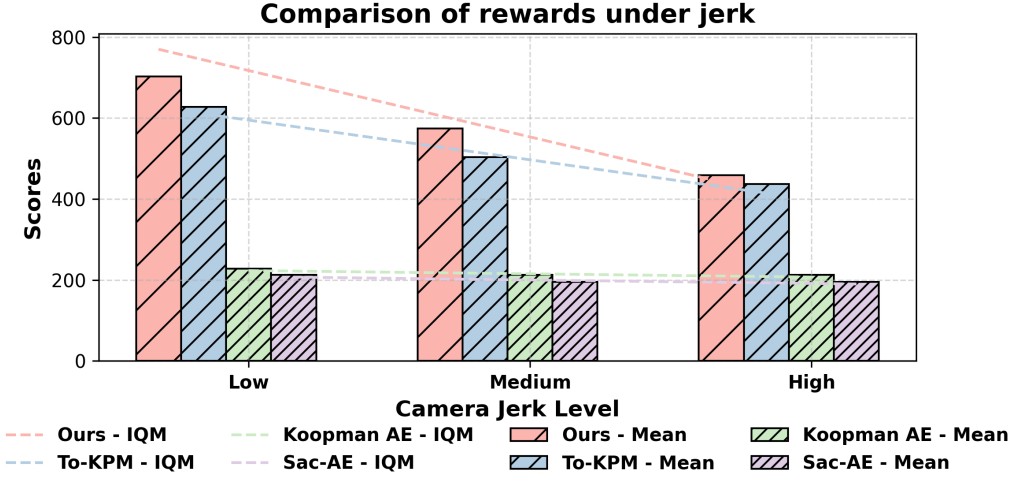

Figure 8: Performance of models with camera jerk

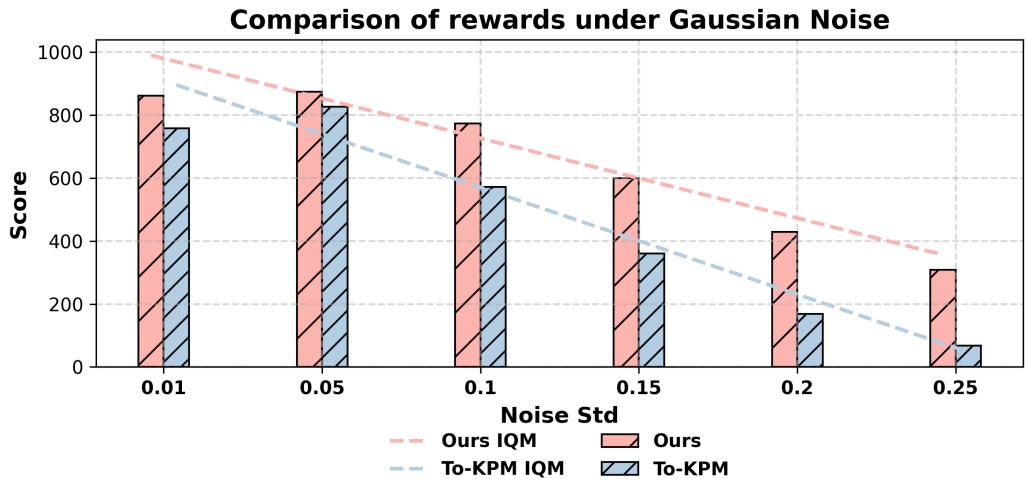

Figure 9: Performance of Models under Gaussian Noise

### D.3 Ablation study on Performance under Imperfect Sensing

This section delves into the resilience of our model when faced with imperfect sensing conditions during evaluation. We specifically examine its performance in two challenging scenarios: *a.) Rainy Environment with Structured Noise*: Unlike Gaussian noise, rain noise[52] presents a more structured and challenging interference, making it difficult for common denoising techniques to effectively mitigate[53][54]. We evaluate the model's performance under three distinct levels of rain density: Low (0.03), Medium (0.75), and High (0.0125). The comparison encompasses predictive models equipped with dynamic predictors, and the outcomes are depicted in Figure 7. The results indicate a general degradation in control performance across all models under test, except for ours, which notably excels by achieving a reward of approximately 400. This demonstrates our model's superior capability to maintain effective system control even in the presence of high noise levels. *b.) Imperfect Sensing due to Camera Jerk*: To further assess our model's robustness, we intro-

duce random camera jerks into the video input stream, simulating real-world sensing imperfections. Three levels of camera jerk are considered: Low jerk (SSIM > 0.8), Medium Jerk (SSIM between 0.4 and 0.5), and High Jerk (SSIM < 0.3). Our findings from Figure 8 reveal that our model consistently outperforms the others under these conditions as well. However, it's noteworthy that the performance gap between our model and the To-KPM model narrows as the jerk intensity increases, with both models exhibiting similar performance metrics at higher jerk levels. Conversely, methods based on autoencoders demonstrate significantly lower performance across all jerk conditions. These evaluations underscore our model's robustness and adaptability to imperfect sensing scenarios, highlighting its potential for real-world applications where sensing conditions are often less than ideal.

### D.4 Ablation Study on Performance under Gaussian Noise

In this experiment, we analyze the robustness of our model's control performance under the influence of Gaussian noise. We introduce zero-mean Gaussian noise to the input images with increasing standard deviation. Figure 9 illustrates the comparative performance of our model and to-kpm [16] model in the presence of Gaussian noise. We exclude models that under performed significantly from this figure, as their performance was too low to be meaningful. Notably, our model demonstrates exceptional resilience, maintaining high performance even under substantial Gaussian noise in the visual input.

