# OpenReview forum: "RoboKoop: Efficient Control Conditioned Representations from Visual Input in Robotics using Koopman Operator"
_robot-learning.org/CoRL/2024/Conference — CoRL 2024_

### Official Review · Reviewer_5VsF · 2024-07-20
**Robotic task verifications are required.**

**Originality:** 3
**Technical Quality:** 3
**Clarity Of Presentation:** 4
**Potential Impact:** 2
**Recommendation:** 2
**Confidence:** 3

**Review:**

The paper's description is clear, and the proposed method's originality is compared with other methods. The paper is of sufficient quality, as the proposed method's effectiveness is verified theoretically and experimentally. On the other hand, the paper's significance could be more questionable, as there are no benchmarks specific to the robotics field, and it is unclear whether the assumptions in the proof of the theorem are held in a real environment.

Strengths
1. A Contrastive Spectral Koopman Embedding network is proposed, which achieves high computational efficiency and agility to disturbances by explicitly representing the linear system introduced by Koopman Theory to a diagonal canonical form.
1. The convergence of the proposed method's policy in the linearized LQR problem on Koopman operators is proved.
1. Computational efficiency and high performance were demonstrated through standard image-based RL benchmarks.

Weakness
1. It has been evaluated only on standard RL benchmarks in the physics simulator and is unsuitable for robotics verification.
The state-space representation is restricted to a diagonal canonical system, which improves learning and computational efficiency. However, based on current experimental results, judging whether this will work on a real robot is difficult.

Thank you for the author's response.
The authors answered my equation and thoughtfully discussed my concerns. However, my recommendation remains unchanged. The proposed idea of explicit diagonal structure in state space is interesting, and the benchmark score is promising for high-dimensional observation. This research progress is expected by conducting further robotics experiments.

**Quality Of The Limitations Section:**

1

**Questions For Rebuttal:**

1. Can the Elapsed time be evaluated about Figure 3? In real robots, the overall computation time, including overhead, needs to be evaluated rather than the number of computational operations.
1. Can the robustness of the physics parameter to disturbances be evaluated about Figure 4?
In robotics applications, the robustness of Sim2Real against environmental changes is also subject to evaluation.
1. Does the assumption hold in a real system, the continuous Objective functions assumed for the proof of Theorem 4?
According to the motivation for which Fazel et al. (2018) derived Lemma 6, it is non-trivial to derive a proof of convergence based on the gradient method for even simple LQR by continuity of the Objective function since the Feedback System (A-BK) can be unstable. Does the proposed method address this point? And did such instability occur in the data of the actual learning process?

Fazel, M., Ge, R., Kakade, S. &; Mesbahi, M.. (2018). Global Convergence of Policy Gradient Methods for the Linear Quadratic Regulator. Proceedings of the 35th International Conference on Machine Learning

**Robotics Focus:**

2

**Summary Of Paper:**

Robot control based on image observation is a challenging problem regarding learning efficiency and robustness. This study proposes a Contrastive Spectral Koopman Embedding network that achieves high computational efficiency and agility to disturbances by explicitly representing the linear state space introduced by Koopman Theory to a diagonal canonical form.  This paper also proves the convergence of the policy learning in the LQR problem on linearized latent spaces based on Koopman operators. It demonstrates its computational efficiency and high performance through standard image-based RL benchmarks.

**Summary Of Recommendation:**

Please see above

---

### Official Review · Reviewer_eUws · 2024-07-24

**Originality:** 3
**Technical Quality:** 3
**Clarity Of Presentation:** 3
**Potential Impact:** 3
**Recommendation:** 3
**Confidence:** 4

**Review:**

Strengths:
1. The paper is theoretically sound and proposes a principled algorithm for learning better visual representations for visual RL

Weaknesses:
1. Robokoop is only evaluated on Deepmind Control Suite; it's unclear whether RoboKoop is applicable to more realistic robot manipulation tasks in either simulation or the real world.

2. Some missing prior works on applying Koopman operator for high-dimensional control, e.g., https://arxiv.org/abs/2303.13446

Additional Questions:

1. I wonder whether RoboKoop could be used in an imitation learning setting? Visual RL may be difficult to apply in the real-world, and the DeepMind control suite tasks are very far away from manipulation tasks in the real world. This paper can be made much compelling if results on manipulation benchmarks, such as MetaWorld or RoboSuite, in either imitation learning/RL setup. A recent work applies Koopman operator to dexterous imitation learning: https://arxiv.org/abs/2303.13446

2. Relatedly, in a lot of manipulation tasks, a single camera observation is not enough to capture all relevant visual information. How to extend RoboKoop to multi-view visual control settings?

3. RoboKoop seems general enough to be used with other RL algorithms. It would be good to show some results on alternative RL algorithms to SAC; this helps demonstrating RoboKoop's generality.

4. Some background on Koopman operator should be provided before introduction of the method.

Overall, I think this is a sound paper that makes sufficient contribution to robot learning; however, its impact may be limited given the relative toy-ish setting of its evaluation.

**Quality Of The Limitations Section:**

1

**Questions For Rebuttal:**

My questions for rebuttal are listed in the review section.

**Robotics Focus:**

2

**Summary Of Paper:**

This paper introduces RoboKoop, a novel contrastive spectral Koopman representations for visual RL; RoboKoop outperforms baseline visual RL algorithms on several DeepMind Control Suite tasks.

**Summary Of Recommendation:**

A nice work on learning linear latent representation for visual RL; however, its applicability to real-world hardware control may be limited.

---

### Official Review · Reviewer_n623 · 2024-07-28
**extension of prior work on using koopman spectral representation**

**Originality:** 2
**Technical Quality:** 3
**Clarity Of Presentation:** 3
**Potential Impact:** 2
**Recommendation:** 2
**Confidence:** 4

**Review:**

Overall, the paper reads like a combination of known techniques to extend previous architectures in order to improve performance as a function of training iterations. The “global linearization” Koopman approach is promising and could better approximate the system dynamics (especially if there are any periodic motions which could otherwise be difficult to encode even with high-dimensional methods). The idea though is not new, as noted below.

The difference from paper: Task-Oriented Koopman-Based Control with Contrastive Encoder from last year’s CoRL should be better clarified. Is the key point that the authors focus on the spectral representation? If so, it is surprising that that paper from last year did not already consider such representation since this is the natural choice to perform the “global linearization”.

The relative impact on performance of the “contrastive” part vs the “spectral” part of the approach is not clear. It would have been useful to show the role of each one through some type of ablation study. Better yet, if the core idea of the paper is the spectral representation, then it would have been clearer and to just focus on that. Otherwise, there are too many complicating details in the paper.

There are key missing references, for instance a Nature Communications paper: Deep learning for universal linear embeddings of nonlinear dynamics, Lusch et al, 2018, which also contains a number of highly relevant references to Koopman representation for nonlinear dynamical systems. Note that the paper by Lusch also explains how to employ a spectral representation using complex eigenvalues.

The video shows two simple examples, and one more complex one – the “Walker”, but it is not clear what the Walker portion is trying to show. Is this the final trained policy (doesn’t look great)?   It would have been more useful to show how a trained policy by the proposed method compares to another competitive existing method at several stages of the training process.

Many references are missing details such as conference or journal names.

**Quality Of The Limitations Section:**

2

**Questions For Rebuttal:**

Explain the impact on performance of contrastive vs spectral part
Explain differences from those prior missing references
Explain novelty and contributions in view of above

**Robotics Focus:**

3

**Summary Of Paper:**

The authors consider control of nonlinear systems through visual scene feedback. They propose to employ spectrally linearized state representation (using Koopman operator) and then RL to perform off-policy control. The claim is that through the choice of complex eigenvalue representation one can achieve sample efficiency through this such task-specific and structure-exploiting representation.

**Summary Of Recommendation:**

The novelty of the contribution is not clear; in addition, the source of improved performance is also not clear.

---

### Official Review · Reviewer_MzWx · 2024-07-31
**More clarifications and experiments may be necessary to make contributions stronger**

**Originality:** 3
**Technical Quality:** 3
**Clarity Of Presentation:** 3
**Potential Impact:** 3
**Recommendation:** 3
**Confidence:** 4

**Review:**

Strength:
1. The Spectral Koopman Embedding network effectively leverages the diagonalized Koopman matrix to enhance the efficiency of Koopman matrix computation and accuracy
in learning task policies.
2.  The paper provides detailed theoretical analysis and proofs of the network's convergence behavior, demonstrating that the task network converges to an optimal policy given sufficient training steps.

Weakness:
1. The experiments conducted are limited in scope, as they only involve 6 of the DM control environments and lack real-world testing. This limitation raises questions about the generalizability and robustness of the conclusions drawn.
2. The explanation of model training in Section 3.2 is somewhat unclear. It seems that the loss function in Equation (3) is used to train both the Koopman matrix parameters ($\lambda_i$) and the encoder parameters. If this is correct, explicitly providing labels about which parameters are being optimized with each of the losses or providing an algorithm overview similar to To-KPM [16] will help reducing ambiguity.
3. The implementation of the spectral Koopman method and the overall algorithm seems incremental, building upon the To-KPM approach [16]. The primary difference introduced in this paper appears to be the diagonalized Koopman matrix, which reduces computational load. However, it is unclear why the convergence theoretical analysis presented cannot be applied to previous work such as To-KPM. The paper lacks a detailed explanation of the specific factors contributing to the observed improvements over To-KPM, which should be addressed to strengthen the findings.

**Quality Of The Limitations Section:**

3

**Questions For Rebuttal:**

1. In Equation (2), the W seems to be introduced without prior context or explanation. Could you please clarify where it originates from?

Also see weaknesses summaries in reviews above.

**Robotics Focus:**

3

**Summary Of Paper:**

The paper presents a new model called the Contrastive Spectral Koopman Embedding network, which is designed to learn efficient linear representations from high-dimensional visual data. This model uses reinforcement learning to perform off-policy control based on the extracted representations, employing a linear controller for this purpose.

**Summary Of Recommendation:**

The paper is mostly clear and provides thorough theoretical and empirical analysis. However, the limited scope of test environments and the similarities between algorithms used in this work and the previous work (specifically To-KPM [16]) makes it appear to be an incremental effort focusing mainly on reducing computation loads. These weaknesses limited the contribution, hence the weak reject recommendation.

---

### Author Rebuttal · Authors · 2024-08-14

Dear Reviewers,

We sincerely appreciate the insightful feedback and valuable suggestions you have provided throughout the review process. Your expertise has greatly contributed to the refinement of our work.

We have uploaded the final ZIP file containing both the main paper and the supplementary materials. Should you have any further questions or require additional clarification, please do not hesitate to let us know.

**If you feel that we have adequately addressed your comments, we kindly request that you consider raising the score of our paper. We believe the revisions have significantly improved the quality of our work and hope this is reflected in your assessment.**

Thank you once again for your invaluable guidance and support.

---

### Decision · Program_Chairs · 2024-09-04

**Decision:**

Accept

**Comment:**

Thank you for your submission to CoRL 2024. Reviewers particularly appreciated the computational and accuracy improvements gained by the proposed spectral Koopman embedding network, the detailed theoretic analysis, and the principled approach.

Reviewers raised concerns about the limited scope of experiments, generalizability, and some missing related work and comparisons. They were unclear about aspects of the training method and concerned that it may be too incremental.

Reviewers valued the authors' detailed and thorough responses, which were sufficient to raise at least one reviewer's initial recommendation. However, concerns linger about generalizability and, in particular, the applicability to real-robot deployment. The reviewers and AC encourage the authors to consider extending experiments to address these lingering concerns for a camera-ready revision.